# Physicochemical Properties and Protein Denaturation of Pork Longissimus Dorsi Muscle Subjected to Six Microwave-Based Thawing Methods

**DOI:** 10.3390/foods9010026

**Published:** 2019-12-25

**Authors:** Ming-Ming Zhu, Ze-Yu Peng, Sen Lu, Hong-Ju He, Zhuang-Li Kang, Han-Jun Ma, Sheng-Ming Zhao, Zheng-Rong Wang

**Affiliations:** 1School of Food Science, Henan Institute of Science and Technology, Xinxiang 453003, China; zerypaul@163.com (Z.-Y.P.); zmm15136790756@163.com (S.L.); hongju_he007@126.com (H.-J.H.); kzlnj1988@163.com (Z.-L.K.); xxhjma@126.com (H.-J.M.); zhaoshengming2008@126.com (S.-M.Z.); wazhro@126.com (Z.-R.W.); 2Henan Province Engineering Technology Research Center of Animal Products Intensive Processing and Quality Safety Control, Henan Institute of Science and Technology, Xinxiang 453003, China; 3National Pork Processing Technology Research and Development Professional Center, Xinxiang 453003, China

**Keywords:** pork *longissimus dorsi* muscle, thawing, physicochemical properties, differential scanning calorimetry, dynamic rheological property

## Abstract

Physicochemical changes and protein denaturation were evaluated for pork *longissimus dorsi* muscle subjected to different thawing methods. Fresh pork *longissimus dorsi* muscle served as a control. Microwave (MT), microwave combined with ultrasonic (MUT), microwave combined with 35 °C water immersion (MIT), microwave combined with 4 °C refrigeration (MRT), microwave combined with air convection (MAT), and microwave combined with running water (MWT) were applied. All microwave-based methods excepted for MT avoided localized overheating. The changes in the water holding capacity (WHC), color, *TBARS*, and protein solubility were lowest with MAT. Differential scanning calorimetry (DSC) and dynamic rheological property measurements indicated, that the MAT samples changed only slightly and presented with complete peaks and high *G′* values compared with the other treatments. Thus, MAT may reduce protein denaturation associated with meat thawing. The results of this study indicated that MAT effectively shortens thawing time, preserves meat quality and uniformity, and could benefit the meat industry and those who consume its products.

## 1. Introduction

In recent years, the meat consumption has grown rapidly because of its better flavor, and nutrition [1]. Though meats are rich in essential amino acids, vitamins, and minerals, they are highly perishable food commodities [2,3]. Frozen storage has been widely used to extend the shelf life and maintain the quality of meat [3]. Frozen meat must be defrosted before consumption or additional processing [4,5]. Frozen meat quality depends on freezing conditions, and thawing methods. Protein denaturation, lipid peroxidation, water loss, textural changes, color and flavor deterioration, and microbial spoilage may occur during thawing [6,7]. Therefore, a suitable thawing method must be considered to maintain quality and minimize losses.

In earlier research studies, traditional methods and newer technologies were used to thaw meat [2,8]. However, air, water, and refrigeration thawing all result in poor meat quality because of prolonged defrosting and wide temperature differences between the external and internal meat layers [9,10]. Microwave, ultrasonics, radio frequency, high-pressure, and ohmics have been proposed and could significantly accelerate meat thawing [11,12]. Nevertheless, each method has its own weaknesses. All of them may cause uneven thawing, protein denaturation, and conformational changes [13]. Hence, it is essential to develop a rapid thawing technique that mitigates undesirable changes in frozen meat and maintains its quality.

Certain researchers proposed and evaluated thawed meat quality by combinations of various methods [14]. The feasibility of this approach has been demonstrated in previous studies [15]. Microwave thawing has been widely used in the meat industry [16]. It can shorten defrosting time and reduce the risk of microbial contamination more effectively than traditional thawing methods [17]. On the other hand, microwave technology has limited practical application in meat defrosting as its penetration is shallow and it may cause localized overheating [18]. Previous studies compared different thawing methods and their effects on freeze-defrost cycles. However, there is little information on the effects of microwave combined with other thawing methods on pork quality. The results of our previous study [19] showed that microwave thawing, was the fastest method and maintained pork quality more effectively than ultrasonic, running water, air, refrigerator or 35 °C water immersion thawing. Nevertheless, microwave thawing caused localized overheating and resulted in poor meat texture. Other thawing methods have various advantages and disadvantages as well. Refrigeration-thawed (RT) meat conserved meat texture and tenderness, while causing protein denaturation, poor WHC, color and thiobarbituric acid-reactive substances (*TBARS*) value. Ultrasonic thawing (UT) was relatively rapid but resulted in poor WHC. 35 °C water immersion thawed (IT) meat conserved meat color, tenderness, and freshness, while other indexes were bad. Running water thawed (WT) meat had better values of *TBARS* and protein solubility, but the DSC indicated the protein denaturation was serious. Air convection thawing (AT) has least impact on WHC. In the present study, then, microwave thawing was combined with ultrasonic, 35 °C water immersion, refrigeration, air convection, and running water and their effects on pork *longissimus dorsi* quality were evaluated. The aim of this study was to identify an optimal microwave-based thawing method that improves thawing efficiency, avoids localized overheating, maintains the pork quality, and weakens disadvantages of the single thawing methods. In this way, a theoretical basis may be established for the development and optimization of a combination thawing process for application in the meat industry.

## 2. Materials and Methods

### 2.1. Sample Preparation

The samples were obtained from five Duroc × Landrace × Yorkshire crossbred pigs selected out of 100 animals slaughtered at Gaojin Food Co. Ltd. (Xinxiang, China). The animals were commercially fattened under intensive rearing conditions to 100 ± 5 kg live weight. Their age range was 170–180 d. At 24 h post-mortem, ten raw *longissimus dorsi* muscles with an average weight of ~2.8–3.0 kg were entirely excised from the right and left sides of each carcass and transported to the laboratory within 30 min. Prior to the experiment, all subcutaneous fat and connective tissues were aseptically removed. Each *longissimus dorsi* muscle was then sliced into four regular loaves (6 cm × 5 cm × 3.5 cm) of equal weight (150 ± 0.5 g). Thirty-five loaves of pork were randomly picked from all 40 samples sections, packaged in polyethylene bags (120 mm × 170 mm) and randomly assigned to seven groups, of which one was a control group, and the others were experimental treatments. Those assigned to the six treatments were immediately stored for 24 h at −20 °C until the subsequent thawing experiments. The control group was immediately analyzed at 20–25 °C.

### 2.2. Thawing Methods

Experimental samples were treated with microwave thawing (MT), microwave combined with ultrasonic thawing (MUT), microwave combined with 35 °C water immersion thawing (MIT), microwave combined with refrigeration (4 °C) thawing (MRT), microwave combined with air convection thawing (MAT), or microwave combined with running water thawing (MWT). All experimental samples were tempered until their core temperature reached 2 °C according to the method of Xia [20] with some modification. Microwave thawing (MT) was performed at 100 W in a household microwave oven (Media Microwave Electronics Co., Ltd., Fushan, China). In a preliminary experiment, various core temperatures (−4 °C, −3 °C, −2 °C, and −1 °C; Figure 1) of the samples thawed with 100 W microwave were measured to determine localized overheating. It was determined that localized overheating occurred when the core temperature was >−4 °C. All combined thawing methods were conducted in a household microwave oven at 100 W until the core temperature reached –4 °C. The second process was the same as that described in Zhu et al. [19]. MUT was run in an ultrasonic cleaner (Ultrasonic Co. Ltd., Kunshan, China) at 100 W and 20 °C. MIT was performed in a water bath (HH 42; Guo Hua Electronics Co. Ltd., Changzhou, China) at 35 °C. For microwave combined with 4 °C refrigeration thawing (MRT), the samples were placed in a refrigerator (Meiling Co. Ltd., Hefei, China) at 4 °C. The samples for MUT, MIT, and MRT were placed in polyethylene bags (120 mm × 170 mm) without holes. MAT was run at 20–25 °C. MAT samples were placed in polyethylene bags (120 mm × 170 mm) perforated on both sides with 16 holes (diameter 6 mm). MWT was conducted at 20–25 °C in water flowing at 0.3 m s^−1^. MWT samples were also placed in polyethylene bags (120 mm × 170 mm) perforated at the bottom with six holes (diameter 6 mm).

### 2.3. Determination of Thawing Time and Rate

Thawing times were determined according to the method of Choi [2] with some modification. Before thawing, holes were made in the sides of the frozen pork samples to enable the thermometers to reach the centers. Testo 160 thermometers (Testo Instruments International Trading Co. Ltd., Shanghai, China) recorded temperature changes in the centers of the samples every minute during thawing until the core temperature reached 2 °C. Other Testo 160 thermometers were also inserted to 0.5 cm depth to record temperature changes at the sample surfaces. According to the method of Zhu et al. [19], the thawing rate was calculated as follows:V_tv_ = L/t(1)
where L is the shortest distance between the sample surface and the center, and t is the time required for the surface and center temperatures to reach 0 °C and 2 °C, respectively.

### 2.4. Determination of Water-Holding Capacity (WHC)

The water-holding capacity (WHC) of thawed pork was expressed in terms of thawing, cooking, drip, centrifugation, and total losses.

Thawing loss was calculated from the weights of the pork samples before (M_0_) and after (M_T_) thawing [21]:Thawing loss (%) = (M_0_ − M_T_)/M_0_ × 100(2)

Cooking loss was determined according to the method of Xia [20] with some modification. Briefly, 10 ± 0.5 g fresh or thawed subsamples were placed in retort pouches and cooked in a water bath at 80 °C until the center temperature reached 70 °C. A Testo 160 thermometer monitored the center temperature. Cooking loss was determined by weighing the samples before (M_0_) and after (M_c_) cooking:Cooking loss (%) = (M_0_ − M_C_)/M_0_ × 100(3)

Centrifugation loss was evaluated by the method of Zhou [22] with slight modification. Fresh or thawed samples (10 ± 0.5 g) were wrapped in filter paper, placed in a centrifuge tube, and centrifuged at 5000 rpm and 4 °C for 10 min. Centrifugation loss was determined by weighing the samples before (M_0_) and after (M_1_) centrifugation:Centrifugation loss (%) = (M_0_ − M_1_)/M_0_ × 100(4)

Drip loss was measured according to the method of Adeyemi [23]. Fresh or thawed samples (10 ± 0.5 g) were weighed and the values were recorded as M_0_. The samples were placed in polyethylene bags, suspended at 4 °C for 24 h, and reweighed. The new values were recorded as M_D_. The drip loss was calculated as follows:Drip loss (%) = (M_0_ − M_D_)/M_0_ × 100(5)

Total loss was determined as follows:Total loss (%) = Thawing loss (%) + Cooking loss (%) + Centrifugation loss (%) + Drip loss (%)(6)

### 2.5. Determination of Shear Force

Shear force was determined for the cooking loss samples according to the method of Li [15]. After the cooking loss calculation, each sample was cut into a cuboid (1 cm × 1 cm × 2 cm). Each cube was perpendicularly sheared in the direction of the muscle fibers using a digital muscle tenderness instrument (C-LM4; Northeast Agricultural University, Harbin, China). The maximum shear force was recorded and expressed in Newtons (N).

### 2.6. Color Determination

According to the method of Chun [24], the surface colors of the fresh and thawed samples were analyzed for *L** (lightness), *a** (redness), and *b** (yellowness) with a color difference meter (CR 400; Konica Minolta Co. Ltd., Tokyo, Japan; standard observer: ~2° observation angle; illuminant: C; aperture: 8 mm). The instrument was precalibrated with a standard white plate. The control *L**, *a**, and *b** values were 97.22, −0.14, and 1.82, respectively. The colorimeter was placed vertically at six different positions on the experimental samples and *L**, *a**, and *b** were recorded. To compare the color values of thawed and fresh samples, total color differences (∆*E**) were calculated and expressed as follows:(7)$$∆E*=\sqrt{{\left(∆L^{*}\right)}^{2}+{\left(∆a^{*}\right)}^{2}+{\left(∆b^{*}\right)}^{2}}$$
where ∆*L**, ∆*a**, and ∆*b** are the differences between the thawed and fresh samples in terms of *L**, *a**, and *b**, respectively.

### 2.7. Thiobarbituric Acid-Reactive Substances (TBARS)

Lipid peroxidation was determined using *TBARS* according to the method described by Xia [20] with slight modifications. Ten grams of each pork sample was weighed out and homogenized at 7500 rpm and 25 °C for 15 s (T25; IKA Works, Inc., Wilmington, NC, USA). Then, 50 mL of 7.5% (*w*/*v*) trichloroacetic acid (TCA) solution was added and the mixture was vortexed for 30 min. The sample solution was filtered through Whatman No.1 filter paper, and then 5 mL of 20 mM 2-thiobarbituric acid was added and the mixture was boiled in a water bath for 40 min. The sample was cooled to 20–25 °C for 30 min and centrifuged at 5500 rpm and 25 °C for 25 min. The absorbance of the supernatant was measured at 532 nm. The *TBARS* values were expressed as mg malondialdehyde (MDA) kg^−1^ sample and calculated as follows:*TBARS* (mg kg^−1^) = (A_532_/W_s_) × 9.48(8)
where A_532_ is the absorbance of the assay solution at 532 nm, W_s_ is the pork sample weight (g), and 9.48 is a constant derived from the dilution factor and the molar extinction coefficient (152,000 L mol^−1^ cm^−1^) of the red TBA product.

### 2.8. Sample Freshness

The pH was measured with a digital pH meter (Seven Compact^TM^; Mettler Toledo, Shanghai, China) according to the method of Zhu [19]. Before measurement, the pH meter was calibrated with pH 6.8 and 4.0 technical buffers according to the instrument instructions. Meat samples (5 g) were homogenized in 45 mL distilled water at 7500 rpm and 25 °C for 15 s and their pH values were measured and recorded.

Total volatile base nitrogen (TVB-N) composes mainly trimethylamine, dimethylamine, ammonia, and other compounds. TVB-N is the product of microbial enzymes that degrade of proteins and nonprotein nitrogenous compounds [1]. For TVB-N analysis, 20 g of each fresh or thawed samples was oscillated for 30 min with 100 mL of 20 g L^−1^ TCA according to the method of Choi [2]. The solution was filtered through Whatman No.1 filter paper, and 5 mL was transferred to a Kjeldahl flask containing 5 mL of 10 g L^−1^ MgO solution. A few drops of dimethicone defoamer were added. The mixture was distilled in 10 mL of 20 g L^−1^ boric acid for 5 min and titrated with 10 mM HCl and methyl red indicator. The TVB-N concentration was evaluated as follows:TVB-N (mg 100 g^−1^) = (V_1_ − V_2_) × c × 14 × 100/(m × 0.05)(9)
where V_1_ and V_2_ are the titration volumes (mL) of the sample and blank solution, respectively, C is the concentration (M) of the HCl solution, and m is the sample weight (g).

Electrical conductivity was measured with a conductivity meter (Seven Compact^TM^; Mettler Toledo, Shanghai, China) according to the method of Yang [25]. In brief, 10 g of each sample was weighed out and homogenized in 100 mL distilled water. The conductivity electrode was then inserted and held in place until a stable reading was obtained.

For the total viable counts, 25 g of each meat sample was weighed out, homogenized for 5 min, and diluted with 250 mL sterile saline (0.85% *w*/*v* NaCl). Then, 1 mL of each solution was spread onto Luria-Bertani (LB) agar (Haibo Biology Co. Ltd., Qingdao, China) plates and incubated at 37 °C for 48 h. Total viable counts were then recorded [26].

### 2.9. Protein Solubility

The effects of thawing on protein solubility were determined according to the method of Joo [27]. Total protein was extracted from 0.25 g pork sample with 5 mL of 0.1 M potassium phosphate buffer (pH 7.2) plus 1.1 M potassium iodide for total protein solubility determination. The samples were minced, homogenized, and placed on a shaker incubator (Fuma Co. Ltd., Shanghai, China) at 4 °C for 12 h. They were then centrifuged at 1500× *g* and 4 °C for 20 min and the protein concentrations in the supernatants were measured with a total protein quantitative assay kit (Nanjing Jiancheng Bioengineering Institute, Nanjing, China). Sarcoplasmic protein was extracted from 0.25 g pork sample using 5 mL of 0.1 M potassium phosphate buffer (pH 7.2). The subsequent steps were the same as those for total protein solubility determination. Differences between total and sarcoplasmic protein solubility were expressed as myofibrillar protein solubility.

### 2.10. Differential Scanning Calorimetry (DSC)

DSC was performed with a STA 449c differential scanning calorimeter (Netzsch Group, Selb, Germany) according to the method of Zhu [19]. The instrument was calibrated for temperature with water and indium and for enthalpy with indium. Fresh or thawed samples weighing 15–20 mg ± 0.01 mg were sealed in Perkin-Elmer sample pans and scanned over a temperature range of 25–100 °C at a heating rate of 5 °C. The thermodynamic values (△H) and the denaturation temperatures (°C) were obtained.

### 2.11. Dynamic Rheological Properties

Rheological differences between the thawed and fresh samples were characterized with a Haake Mars 40 Rheometer (Thermo Fisher Scientific, Waltham, MA, USA) using cone-plate geometry (0.5 mm gap). Measurements were made at 20 °C over 10 min and a heating rate of 2 °C min^−1^ over a range of 20−90 °C. A constant strain amplitude of 10% was selected to cut the samples at 0.1 Hz. Variations in the storage modulus (*G’*) with temperature were recorded [28].

### 2.12. Statistical Analysis

The carcass was the experimental unit. There were five replicates and the data were means ± SEM with significant differences at *p* < 0.05. Variances were determined and Duncan’s multiple range tests were performed in SPSS v. 18.1 (IBM Corp., Armonk, NY, USA).

## 3. Results and Discussion

### 3.1. Thawing Time and Rate

Temperature changes with time for frozen samples subjected to six different thawing methods are shown in Figure 2. The rate-limiting step in the thawing process was passage from the region of maximum ice crystal formation (−5–−1 °C). Thereafter, thawing accelerated. This pattern is consistent with the food freezing. MT had the shortest thawing time (4 min) followed by MIT (20 min), MUT (25 min), and MWT (30 min). MRT had the longest thawing time (960 min) followed by MAT (110 min). These discrepancies may be explained by the relative differences in heat transfer rate among the samples exposed to ultrasonic, water, air convection, and 4 °C refrigeration thawing [2]. Nevertheless, microwave combined with the other thawing methods shortened the thawing time compared with those obtained using each individual technique. According to Zhu [19], thawing time could be reduced by ~50% using MAT instead of air convection thawing (AT). The highest thawing rate was recorded for MT (39.61 cm h^−1^) followed by MUT (9.18 cm h^−1^), MIT (8.97 cm h^−1^), MWT (7.93 cm h^−1^), MAT (2.24 cm h^−1^), and MRT (0.446 cm h^−1^). Our previous study [19] showed the thawing rate of UT, IT, WT, AT, and RT was 6.22 cm h^−1^, 4.34 cm h^−1^, 3.20 cm h^−1^, 0.95 cm h^−1^, and 0.125 cm h^−1^, respectively. Therefore, the thawing rates measured here were significantly higher for the combined than the single methods (*p* < 0.05).

### 3.2. Effect of Thawing on WHC

Water loss may adversely affect meat weight, appearance, color, and sensory properties as it is associated with the loss of certain amino acids and nucleotides [20,29]. Table 1 shows the WHC of the experimental samples subjected to thawing, cooking, drip, and centrifugation and the total loss under various thawing methods. Relative to the control, all methods caused significant thawing loss (*p* < 0.05). The highest thawing loss was recorded for samples treated with MRT (5.37%) because thawing by this method was too slow. Ambrosiadis [30] indicated that for beef, slow defrosting caused substantially greater thawing loss than microwaving. The lowest value (1.74%) was obtained for the samples treated with MWT. It was similar to that reported for running water thawing (1.86%) [19] but still comparatively lower than it. Microwaving combined with other methods could reduce thawing loss. Leygonie [3] reported that accelerated thawing could cause thawing loss-type damage, associated with meat tissues. However, short-term microwave thawing causes relatively higher losses as it heats product rapidly and accelerates evaporation. The phenomenon was observed in pork by Kondratowicz [31], in beef by Kim [32], in chicken by Anna [33], and in rabbit meat by Chwastowska-Siwiecka [34]. Therefore, thawing loss tended to increase with thawing temperature.

Centrifugation, cooking, and drip losses for the MAT treatment were 20.84%, 21.23%, and 4.76%, respectively, and most nearly approached those of the control (20.36%, 21.77%, and 4.50%, respectively). However, they were significantly lower (*p* < 0.05) than those for the other thawing treatments. The total loss for the MAT treatment (49.26%) was lowest of all, and was, in fact, lower than those previously reported for either MT or AT alone [19]. Moisture loss occurred when frozen muscle tissue was thawed, and resulted in higher centrifugation, cooking, and drip losses than those of fresh muscle tissue [35]. However, another study reported no significant differences between thawed and fresh samples [8]. Changes in muscle moisture content were affected by freezing and thawing [3]. Hence, the relative differences in water loss among the various thawing treatments may be explained by the relative differences in the samples, freezing and thawing rates, and final thawing temperatures [2].

### 3.3. Effect of Thawing on Shear Force

Tenderness is an important criterion of pork quality. It is assessed by measuring shear force (N). Meat tenderness increases with decreasing shear force. The effects of the various thawing treatments on shear force are shown in Table 1. The range of shear force obtained for pork samples subjected to the various thawing methods was 27.34–35.98 N. The lowest N was determined for the MRT samples. Xia [20] reported that thawing temperature influences pork *longissimus dorsi* muscle tenderness. They measured comparatively lower N for samples thawed at lower temperatures. Except for MWT, the values of N for all thawed samples approached that of the control. Elevated temperature and extended thawing time may denature or degrade protein [15]. The shear forces of the MAT samples resembled those of the fresh ones. MAT decreased the shear force of defrosted pork by 16.2% relative to that for AT-thawed meat [19]. Thus, combining thawing methods could improve meat tenderness in the thawing process. This hypothesis is consistent with the report of Zhang [14] and Li [15].

### 3.4. Effect of Thawing on Pork Color

Color changes in meat during frozen storage and tempering is used to assess meat quality. As shown in Table 2, *L** for MRT-thawed meat was significantly lower than that for fresh meat (*p* < 0.05). In contrast, *L** for the samples subjected to MT, MUT, MIT, MAT, and MWT did not significantly differ from that for the fresh control (*p* > 0.05). This observation is similar to those reported by Anna [33] and Benli [36]. These authors demonstrated that thawing method had no significant influence on *L**. Hughes [37] found that decreases in the meat water-holding capacity reduced surface light reflectivity which, in turn, lowered *L**. All thawing methods except for MAT significantly decreased *a** relative to the fresh control (*p* < 0.05). The value of *a** changes rapidly during thawing possibly because of lipid and protein oxidation, microstructural changes, and drip loss [38]. Microwave thawing may reduce both protein denaturation and loss of quality in frozen meat [33]. Conversely, Choi [2] reported substantial changes in the color values of microwave-thawed pork loin caused by non-uniform heating. Here, no significant differences in *b** were detected between the experimental groups and the control (*p* > 0.05). The total color difference (∆*E**) was also evaluated. The lowest ∆*E** (1.18) was obtained for samples thawed by MAT. It was lower than that measured for AT-thawed meat [19]. Therefore, MAT thawing maintained color stability in thawed pork. 

### 3.5. Effect of Thawing on TBARS

The effects of thawing methods on *TBARS* were also evaluated and the results are shown in Table 2. The *TBARS* values for MT and MAT were similar to those for the control (0.16 mg kg^−1^). On the other hand, significantly higher *TBARS* values were measured for the samples thawed by MUT, MIT, MRT, and MWT (*p* < 0.05). Previous studies reported that under freezing conditions, frozen storage, and thawing could cause the autooxidation of polyunsaturated fats [39]. Chun [24] found that the thawing conditions had a much greater influence on *TBARS* than the freezing conditions. Thus, the comparatively higher *TBARS* values measured for MUT, MIT, and MWT may be explained by their higher thawing temperatures and microbial action. The *TBARS* values for all thawing samples did not exceed the flavor threshold (>1.0 mg kg^−1^), above which meat product odor and flavor are unacceptable [40].

### 3.6. Effect of Thawing on Freshness

The impact of thawing on freshness was assessed to determine its direct relationship with edibility and safety. Here, freshness was evaluated by pH, total volatile base nitrogen (TVB-N), electrical conductivity, and total viable counts (Figure 3). The pH is an objective meat quality parameter. It changes in response to the various phases of water transformation and declines during thawing as acidic by-products accumulate [33]. Leygonie [3] stated that inappropriate thawing actually increases pH. As shown in Figure 3a, however, there were negligible differences in pH among the various thawed samples (*p* > 0.05). Similar results were reported by Zhu [19] for pork, by Zhang [14] for chicken, and by Chwastowska-Siwiecka [34] for rabbit meat. The pH range was 5.82–5.92. Therefore, a high level of freshness was achieved after the six thawing treatments. TVB-N is an important index of meat freshness. The highest acceptable TVB-N was 20 mg N 100 g^−1^ [8]. Here, the TVB-N of the control was 6.05 mg N 100 g^−1^ (Figure 3b). Thus, the TVB-N of fresh pork was generally within the acceptable range. The TVB-N range for the pork samples subjected to the six thawing treatments was 6.8–8.1 mg N 100 g^−1^ (*p* < 0.05). The MAT samples had the lowest TVB-N (6.8 mg N 100 g^−1^). Nevertheless, all TVB-N were <20 mg N 100 g^−1^ which implies that a high standard of freshness was maintained after freezing and thawing. This observation was consistent with that reported by Choi [2]. Electrical conductivity (EC) is another important meat freshness indicator. The electrical conductivity of pork is negatively correlated with freshness. As the meat tissue degrades, it produces large quantities of conductive substances. The electrical conductivity of the control was 1284.67 μS cm^−1^ (Figure 3c). The electrical conductivity range for the thawed frozen pork samples was 1336.83–1369.33 μS cm^−1^ (*p* < 0.05). The upper threshold of acceptance is 1370 μS cm^−1^ [25]. There were no significant differences among treatments in terms of EC. The electrical conductivity of the pork was positively correlated with TVB-N. Yang [25] reported similar findings. Favorable temperatures and long processing times during thawing may increase microbial growth [3]. Here, the total viable counts in the control were 1.46 lg CFU g^−1^ (Figure 3d). It was comparatively higher for the samples after the various thawing treatments. The MT samples had a relatively low total viable counts (1.68 lg CFU g^−1^) because of its short processing time and transient high temperature. The MAT samples presented with the lowest total viable counts (1.80 lg CFU g^−1^) which was significantly lower than that for the AT sample (2.91 lg CFU g^−1^) (*p* < 0.05) [19]. Therefore, the combined MAT thawing method may effectively reduce microbial growth. Overall, it was apparently the best for maintaining pork freshness.

### 3.7. Effect of Thawing on Protein Solubility

Protein solubility is a vital metric of meat quality and is closely associated with several other physical and functional characteristics [41]. Here, the influences of thawing methods on the three types of protein solubility were investigated (Figure 4). Compared with the control, the thawed samples had significantly lower values for all three types of protein solubility (*p* < 0.05). Decrease in protein solubility is a marker of muscle protein deterioration and is associated with increases in surface hydrophobicity and exudation [14]. Freezing and thawing may reduce pork protein solubility. Pork samples processed by MAT and MWT had significantly higher total and myofibrillar protein solubilities than those treated with the other thawing methods (*p* < 0.05). Nevertheless, there were no significant differences among thawing treatments in terms of sarcoplasmic protein solubility (*p* > 0.05). A previous study showed that insoluble protein aggregation may occur, and reduce myofibrillar protein solubility and extractability [41]. In the current study, MAT maintained significantly higher protein solubility than either MT or AT (*p* < 0.05) [19].

### 3.8. Effects of Thawing on Protein Denaturation as Determined by Differential Scanning Calorimetry (DSC)

Thermal transition curves for samples subjected to various thawing methods are shown in Figure 5. For the control, three major endothermic transitions occurred. The first occurred at peak temperatures between 53 °C and 58 °C and corresponded to myosin denaturation. The second happened between 64 °C and 67 °C when collagen and sarcoplasmic proteins denatured. The third was observed between 73 °C and 77 °C wherein actin was denatured. Similar results were reported by Ali [42]. Each thawing method produced different protein denaturation peaks. Peak 2 was significantly lower for the MIT samples compared with the control. Thus, MIT denatured collagen and sarcoplasmic proteins. Table 3 shows the denaturation temperature T_m_ (°C) and enthalpies ∆H (J g^−1^) of pork muscle protein subjected to various thawing conditions and analyzed by DSC. There was no significant difference between Peaks 1 and 2 in terms of T_m_ (*p* > 0.05). The Peak 3 T_m_ for MT, MUT, and MWT were higher than that of the control (*p* < 0.05). The enthalpy ∆H (J g^−1^) for Peak 2 was lowest for pork samples subjected to MIT. This finding aligned with the results shown in Figure 5 and resembled those reported by He [8]. Water immersion thawing at 50 °C significantly influenced protein denaturation whereas the AT method had the least effect on it. Only small changes in protein denaturation were observed for the MAT samples with complete peaks and high ∆H values. Thus, MAT only slightly induced protein denaturation. This discovery is consistent with the fact that MAT presented with the lowest total water loss rate.

### 3.9. Effects of Thawing on Dynamic Rheological Properties

A dynamic rheological test was conducted to assess heat-induced myofibrillar proteins gelation. This parameter reflects protein quality [28]. *G*’ is a measure of the deformation energy stored in the pork sample during shearing and indicates the elastic behavior of the material [42]. The changes in *G*’ for the pork samples subjected to various thawing methods are shown in Figure 6. For the control, *G*’ gradually decreased from 20 °C to 44 °C and then slightly increased from 45 °C to 50 °C when the protein underwent denaturation and gelation [42]. *G*’ moderately declined thereafter until 55 °C when the myosin tails denatured. *G*’ then rapidly increased up to 80 °C when the viscous sol was transformed into an elastic gel network [43]. After thawing, the pattern of change in *G*′ was similar to that for the control but *G*’ (40,206–56,655 Pa) was lower than that of the control (~59,796 Pa). Thawing had a deleterious effect on the dynamic rheological properties because it induced denaturation, excessive aggregation, and structural changes in the proteins [42]. *G*’ was highest (56,655 Pa) for the samples subjected to MAT and was similar to that for the control. Therefore, MAT minimizes protein denaturation and structural changes during heating. These findings aligned with the results obtained for DSC. *G*’ in the present study was higher than the loss modulus (*G*”). Thus, the material behaved like a solid and its deformations were essentially elastic [44]. Other physicochemical changes including the oxidation and denaturation of proteins extracted from pork *longissimus dorsi* muscle thawed by MAT should be assessed in the future to elucidate the mechanism of water retention in the process.

## 4. Conclusions

MAT provided the best results in terms of WHC, color, *TBARS*, protein solubility, and protein denaturation. There were no significant differences among the six thawing treatments in terms of pH, TVB-N, or electrical conductivity. Microwave-based combination thawing avoided localized overheating and maintained the uniformity of the frozen pork samples. Therefore, MAT was the preferred thawing method as it maintained sample uniformity and quality. MAT is a promising way to improve pork meat quality during freezing and thawing and could be beneficial both to the frozen meat industry and to those who consume its products.

## Figures and Tables

**Figure 1 foods-09-00026-f001:**
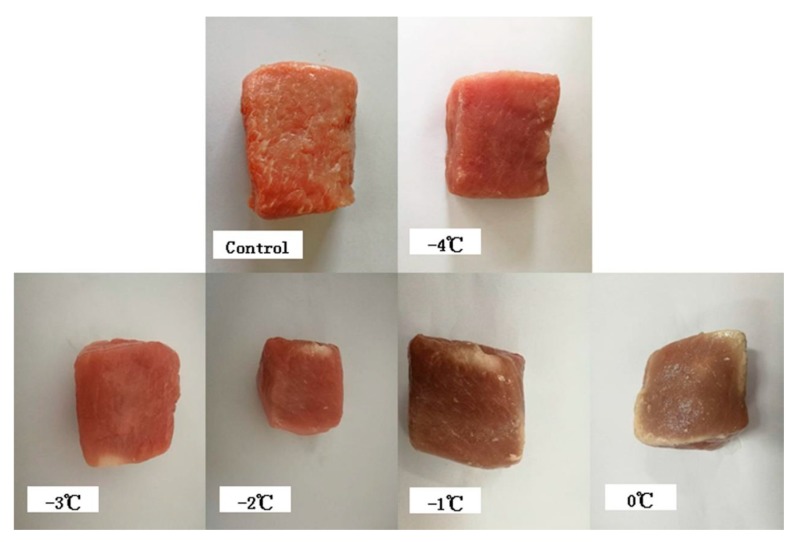
Localized overheating in pork samples thawed by microwave. Core temperatures were −4 °C, −3 °C, −2 °C, and −1 °C.

**Figure 2 foods-09-00026-f002:**
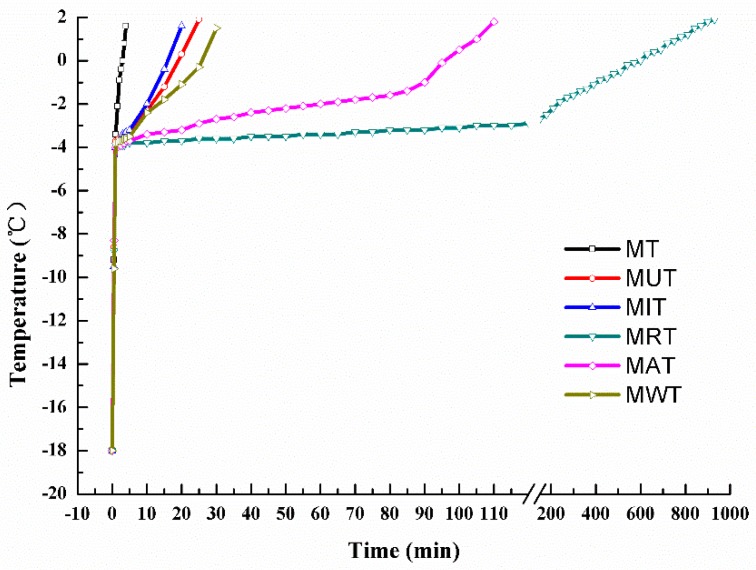
Temperature changes in frozen pork samples under six different thawing conditions. MT: Microwave thawing (100 W); MUT: Microwave combined with ultrasonic thawing; MIT: Microwave combined with 35 °C water immersion thawing; MRT: Microwave combined with 4 °C refrigeration thawing; MAT: Microwave combined with air convection thawing; MWT: Microwave combined with running water thawing.

**Figure 3 foods-09-00026-f003:**
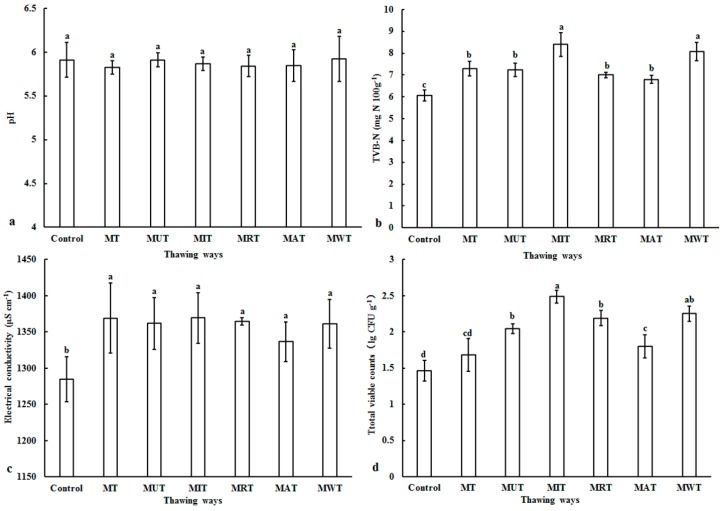
Changes in pH (**a**), total volatile base nitrogen (TVB-N, (**b**)), electrical conductivity, (**c**) and total viable counts (**d**) of pork samples under six different thawing conditions. MT: Microwave thawing (100 W); MUT: Microwave combined with ultrasonic thawing; MIT: Microwave combined with 35 °C water immersion thawing; MRT: Microwave combined with 4 °C refrigeration thawing; MAT: Microwave combined with air convection thawing; MWT: Microwave combined with running water thawing. ^a–d^ values in the same series followed by different letters are significantly different according to Duncan’s multiple range test (*p* < 0.05).

**Figure 4 foods-09-00026-f004:**
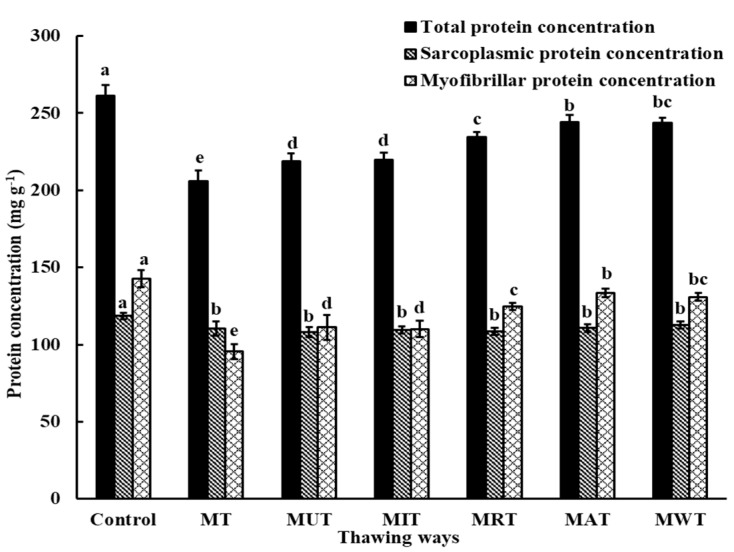
Changes in pork protein solubility under six different thawing conditions. MT: Microwave thawing (100 W); MUT: Microwave combined with ultrasonic thawing; MIT: Microwave combined with 35 °C water immersion thawing; MRT: Microwave combined with 4 °C refrigeration thawing; MAT: Microwave combined with air convection thawing; MWT: Microwave combined with running water thawing. ^a–d^ values in the same series followed by different letters are significantly different according to Duncan’s multiple range test (*p* < 0.05).

**Figure 5 foods-09-00026-f005:**
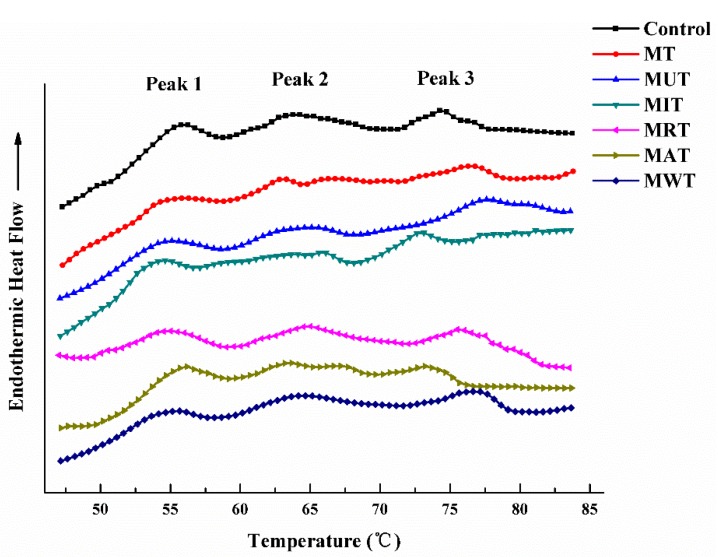
Typical differential scanning calorimetry (DSC) scans of the muscle proteins in pork under six different thawing conditions. MT: Microwave thawing (100 W); MUT: Microwave combined with ultrasonic thawing; MIT: Microwave combined with 35 °C water immersion thawing; MRT: Microwave combined with 4 °C refrigeration thawing; MAT: Microwave combined with air convection thawing; MWT: Microwave combined with running water thawing.

**Figure 6 foods-09-00026-f006:**
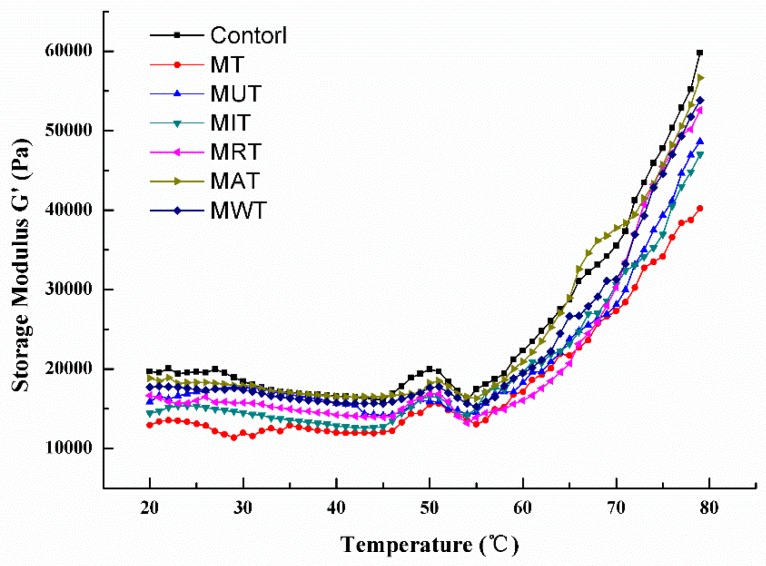
Changes in dynamic storage modulus (*G’* Pa) of pork samples under six different thawing conditions. MT: Microwave thawing (100 W); MUT: Microwave combined with ultrasonic thawing; MIT: Microwave combined with 35 °C water immersion thawing; MRT: Microwave combined with 4 °C refrigeration thawing; MAT: Microwave combined with air convection thawing; MWT: Microwave combined with running water thawing.

**Table 1 foods-09-00026-t001:** Changes in water holding capacity of pork samples under six different thawing conditions.

Parameter	Thawing Ways						
	Control	MT	MUT	MIT	MRT	MAT	MWT
Thawing loss (%)	-	2.26 ± 0.15 ^c^	3.05 ± 0.27 ^b^	3.27 ± 0.20 ^b^	5.34 ± 0.09 ^a^	2.50 ± 0.19 ^c^	1.74 ± 0.12 ^d^
Centrifugation loss (%)	20.69 ± 0.44 ^c^	23.09 ± 0.22 ^a^	23.31 ± 0.37 ^a^	23.59 ± 1.11 ^a^	21.93 ± 0.49 ^b^	20.84 ± 0.16 ^c^	21.93 ± 0.35 ^b^
Cooking loss (%)	21.77 ± 0.20 ^de^	22.46 ± 0.54 ^bcd^	22.36 ± 0.73 ^cd^	22.82 ± 0.19 ^abc^	23.38 ± 0.66 ^ab^	21.23 ± 0.39 ^e^	23.48 ± 0.59 ^a^
Drip loss (%)	4.50 ± 0.16 ^d^	6.53 ± 0.11 ^a^	5.96 ± 0.40 ^bc^	5.66 ± 0.28 ^c^	6.32 ± 0.38 ^ab^	4.76 ± 0.11 ^d^	5.68 ± 0.17 ^c^
Total loss (%)	-	54.35 ± 0.85 ^b^	54.68 ± 0.69 ^b^	55.35 ± 0.84 ^b^	56.97 ± 0.97 ^a^	49.34 ± 0.71 ^d^	52.80 ± 0.96 ^c^
Shear force (N)	31.08 ± 0.96 ^bc^	28.77 ± 1.01 ^de^	29.43 ± 1.20 ^cd^	27.56 ± 0.98 ^de^	27.34 ± 0.83 ^e^	31.71 ± 1.34 ^b^	35.98 ± 1.00 ^a^

MT: Microwave thawing (100 W); MUT: Microwave combined with ultrasonic thawing; MIT: Microwave combined with 35 °C water immersion thawing; MRT: Microwave combined with 4 °C refrigeration thawing; MAT: Microwave combined with air convection thawing; MWT: Microwave combined with running water thawing. Data are means ± SE. ^a–d^ values in the same row followed by different letters are significantly different according to Duncan’s multiple range test (*p* < 0.05).

**Table 2 foods-09-00026-t002:** Changes in color and thiobarbituric acid-reactive substances (*TBARS)* values of pork samples under six different thawing conditions.

Parameter	Thawing Ways						
	Control	MT	MUT	MIT	MRT	MAT	MWT
L*	50.18 ± 1.99 ^ab^	51.94 ± 2.22 ^a^	48.56 ± 0.68 ^bc^	52.70 ± 0.82 ^a^	47.01 ± 0.19 ^c^	51.22 ± 1.97 ^ab^	52.92 ± 1.07 ^a^
a*	6.27 ± 0.09 ^a^	5.99 ± 0.01 ^b^	5.78 ± 0.21 ^bc^	5.98 ± 0.14 ^b^	5.57 ± 0.17 ^c^	6.21 ± 0.03 ^a^	5.56 ± 0.08 ^c^
b*	4.82 ± 0.36 ^ab^	5.24 ± 0. 29 ^a^	4.65 ± 0.05 ^b^	5.11 ± 0.29 ^ab^	4.74 ± 0.18 ^b^	4.93 ± 0.23 ^ab^	4.90 ± 0.18 ^ab^
∆E	-	1.92 ± 0.12 ^c^	2.09 ± 0.24 ^c^	2.64 ± 0.37 ^bc^	3.41 ± 0.23 ^a^	1.18 ± 0.39 ^d^	2.90 ± 0.77 ^ab^
*TBARS* (mg kg^−1^)	0.16 ± 0.01 ^d^	0.17 ± 0.01 ^cd^	0.26 ± 0.02 ^c^	0.23 ± 0.01 ^b^	0.18 ± 0.01 ^c^	0.16 ± 0.01 ^d^	0.27 ± 0.02 ^a^

MT: Microwave thawing (100 W); MUT: Microwave combined with ultrasonic thawing; MIT: Microwave combined with 35 °C water immersion thawing; MRT: Microwave combined with 4 °C refrigeration thawing; MAT: Microwave combined with air convection thawing; MWT: Microwave combined with running water thawing. Data are means ± SE. ^a–d^ values in the same row followed by different letters are significantly different according to Duncan’s multiple range test (*p* < 0.05).

**Table 3 foods-09-00026-t003:** Denaturation temperature T_m_ (°C) and enthalpy △H (J g^−1^) for the muscle proteins in pork samples under six different thawing conditions (analyzed by differential scanning calorimetry (DSC)).

Thawing Ways	Peak 1		Peak 2		Peak 3	
	T_m_ (°C)	△H (J g^−1^)	T_m_ (°C)	△H (J g^−1^)	T_m_ (°C)	△H (J g^−1^)
Control	55.76 ± 1.34 ^a^	0.31 ± 0.01 ^a^	66.26 ± 2.02 ^a^	0.20 ± 0.02 ^a^	74.26 ± 1.10 ^cd^	0.19 ± 0.03 ^ab^
MT	55.29 ± 0.94 ^a^	0.28 ± 0.02 ^bc^	66.80 ± 2.00 ^a^	0.16 ± 0.03 ^b^	76.30 ± 0.55 ^ab^	0.17 ± 0.01 ^bc^
MUT	55.10 ± 0.62 ^a^	0.25 ± 0.02 ^d^	65.10 ± 1.23 ^a^	0.17 ± 0.02 ^ab^	77.60 ± 0.83 ^a^	0.20 ± 0.01 ^ab^
MIT	54.62 ± 0.87 ^a^	0.31 ± 0.01 ^a^	65.62 ± 2.04 ^a^	0.11 ± 0.01 ^c^	73.12 ± 0.77 ^d^	0.16 ± 0.02 ^c^
MRT	55.02 ± 0.93 ^a^	0.26 ± 0.01 ^cd^	65.02 ± 2.05 ^a^	0.19 ± 0.01 ^ab^	75.52 ± 0.63 ^bc^	0.22 ± 0.02 ^a^
MAT	56.19 ± 0.77 ^a^	0.29 ± 0.01 ^ab^	64.19 ± 1.47 ^a^	0.18 ± 0.01 ^ab^	73.19 ± 0.90 ^d^	0.20 ± 0.01 ^ab^
MWT	55.65 ± 0.92 ^a^	0.26 ± 0.01 ^cd^	64.65 ± 1.79 ^a^	0.17 ± 0.01 ^ab^	76.65 ± 1.10 ^ab^	0.22 ± 0.02 ^a^

MT: Microwave thawing (100 W); MUT: Microwave combined with ultrasonic thawing; MIT: Microwave combined with 35 °C water immersion thawing; MRT: Microwave combined with 4 °C refrigeration thawing; MAT: Microwave combined with air convection thawing; MWT: Microwave combined with running water thawing. Data are means ± SE. ^a–d^ values in the same row followed by different letters are significantly different according to Duncan’s multiple range test (*p* < 0.05).

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
