# Peer review of "Physicochemical Properties and Protein Denaturation of Pork Longissimus Dorsi Muscle Subjected to Six Microwave-Based Thawing Methods"

_foods, 2019, doi:10.3390/foods9010026_

Round 1
Reviewer 1 Report
Title:
“longissimus dorsi” should be in italic (check in all the manuscript)
Abstract:
“All microwave-based methods shortened thawing time, and avoided localized overheating”
L19-20 This results are not from the present study; all treatments were microwave-based.
Introduction:
L31-32 “In recent years, the meat industry has grown rapidly because its products have excellent nutritional properties” I do not agree with the sentence
L37 Maybe better to say “flavour deterioration” rather than “flavor loss”
L37 “inevitably occurred” I am not sure if always all these changes are seeing (it will depend on conditions. Maybe better to say: … would probably occur…
L39 “Traditional methods and newer technologies have been used to thaw meat” Do you mean in the industry or in research studies?
L44 “curing”? or “thawing”?
L47-48 “to guarantee frozen meat”?
L57 “based on the previous study” You should explain what was done/concluded in the previous study and what will be addressed in the present study.
You should explain better why did you choose the different treatments used.
Material and method:
L67 “from alternate sides of each carcass” I do not understand this. How many LD were obtained from the carcasses, one or two?, left or right?
Was the entire muscle removed or just a section. If it was a section, please explain localization in the muscle.
L72 Were the five samples from each treatment from different carcasses?
L92 What was the temperature of the running water?
We could better compare running with static water, if it was a the same temperature…
Was the freezing time equal in all treatments?
Figure one. The size of the samples seems to be different between each other.
How did you measure the localized overheating?
2.2. Thawing methods:
The bags had holes… so water was in contact with meat when thawing in some treatments?
L146 I do not see that TBARS was conducted in the reference cited.
L191 DSC was not conducted in the reference cited.
Results and discussion:
3.1. Thawing time and rate
L217 “Overall, the thawing rates were higher for the combined methods than the single methods.”
I suppose authors are right, but there is no data to prove that.
I do not see results of statistical analysis in this section. Did you do it?
In Figure 2, I can see that MAT and MRT seems to accelerate the speed of the thawing rate at the end of the thawing process. Do you think that this is maybe something interesting to mention?
L233-234: …”and resembled those reported for single thawing treatments [19] but were, in fact, still relatively lower than them”
I cannot found the reference, so I do not understand what single thawing treatments are you referring to. You should explain further.
L235 “Leygonie [3] found”
Saying “found” sounds such as it is a research study, but it is a review…
L235 “accelerated thawing reduced damage”
Could you explain further this, please?
Figure 4 “sarcoplasmic”
L348 Could also the sarcoplasmic protein loss in thawing loss influence these results?
Conclusion:
L427-428: “In the present study, microwave in combination with other thawing methods effectively reduced thawing time compared to the individual conventional thawing methods. Thawing time could be cut in half by using MAT instead of AT.”
This is not a conclusion of the present study. Individual conventional thawing methods were not shown.
L434: …” shorter thawing times” Is shorter compared only to one treatment.
Author Response
Dear Reviewer,
Thanks for your decision letter on 12th December 2019 about our manuscript.
Manuscript No.: foods-661457.
Title: Physicochemical properties and protein denaturation of pork longissimus dorsi muscle subjected to six microwave-based thawing methods
We appreciated your valuable suggestion and comments. Based on your suggestions and comments, we have thoroughly revised the manuscript and showed in revision mode.
Detailed revisions and responses to the comments are listed below.
“longissimus dorsi” should be in italic (check in all the manuscript)
Thanks so much for your advice. The ‘longissimus dorsi’ has been revised to ‘longissimus dorsi’ uniformly throughout the manuscript.
Abstract: “All microwave-based methods shortened thawing time, and avoided localized overheating” L19-20 This results are not from the present study; all treatments were microwave-based.
Thanks for your careful review. The sentence has been changed to ‘All microwave-based methods excepted for MT avoided localized overheating’ as shown on line 19-20.
L31-32 “In recent years, the meat industry has grown rapidly because its products have excellent nutritional properties” I do not agree with the sentence.
Thank you very much for the suggestion. The sentence has been rewritten as shown on line 31-32 and the relevant reference [1] has been changed accordingly in the reference part.
L37 Maybe better to say “flavour deterioration” rather than “flavor loss”.
Thank you very much for the suggestion. The ‘flavor loss’ has been revised to ‘flavor deterioration’.
L37 “inevitably occurred” I am not sure if always all these changes are seeing (it will depend on conditions. Maybe better to say: … would probably occur….
Thank you very much for your comments. ‘would be inevitably occurred’ has been changed to ‘may occur’ and shown on line 38.
L39 “Traditional methods and newer technologies have been used to thaw meat” Do you mean in the industry or in research studies?
Thank you very much for your comments. Here, we meant the methods have been used to thaw meat in the research studies and the relevant details were supplemented on line 40-41 to avoid ambiguity.
L44 “curing”? or “thawing”?
Thanks for your careful review. The ‘curing’ has been changed into ‘thawing’ on line 46.
“to guarantee frozen meat”?
Thanks for your careful review. The sentence has been rewritten on line 49-50.
“based on the previous study” You should explain what was done/concluded in the previous study and what will be addressed in the present study. You should explain better why did you choose the different treatments used.
Thanks so much for your suggestion. Five ways were chosen based on our previous study. In our previous study (Effects of rapid and slow thawing methods on quality characteristics and protein denaturation of frozen pork), these thawing ways have been studied. The results showed that microwave thawing, was the fastest method and maintained pork quality more effectively than ultrasonic, running water, air, refrigerator or 35 °C water immersion thawing. Nevertheless, microwave thawing caused localized overheating and resulted in poor meat texture. Other thawing methods have various advantages and disadvantages as well. Refrigeration-thawed (RT) meat conserved meat texture and tenderness, while caused protein denaturation, poor WHC, color and thiobarbituric acid-reactive substances (TBARS) value. Ultrasonic thawing (UT) was relatively rapid but resulted in poor WHC. 35 ℃ water immersion thawed (IT) meat conserved meat color, tenderness and freshness, while other indexes were bad. Running water thawed (WT) meat had better values of TBARS and protein solubility, but the DSC indicated the protein denaturation was serious. Air convection thawing (AT) has least impact on WHC. Moreover, air convection and water thawing are widely applied in the meat industry and home currently. As well, there is little information on the effects of microwave used in combination with other thawing methods on pork quality. Hence, the microwave-based thawing has been studied to shorten thawing time and avoid localized overheating. According to your suggestion, ‘what was done in the previous study’ has been supplemented as shown on line 58-68.
L67 “from alternate sides of each carcass” I do not understand this. How many LD were obtained from the carcasses, one or two? left or right? Was the entire muscle removed or just a section. If it was a section, please explain localization in the muscle.
Thank you very much for the suggestion. Two LD were obtained from the two sides (left and right) of each carcass. The entire longissimus dorsi muscle was excised for our experiment. Therefore, ten raw longissimus dorsi muscles were entirely excised from left and right sides of each carcass in total. The sentence has been rewritten to clarify this point and shown on line 83-84.
L72 Were the five samples from each treatment from different carcasses?
Thanks for your careful review. The five samples from each treatment were randomly allocated from 35 loaves of pork which were picked from all 40 samples randomly. The sentences have been rewritten to avoid confusing readers and shown on line 87-90.
L92 What was the temperature of the running water? We could better compare running with static water, if it was the same temperature…
Thanks so much for your questions. The temperature of the running water was 20 ~ 25°C. It was not a same temperature with static water. We do not account for the same temperature. In the present study, the aim was to select an optimal microwave-based thawing method. Here, the temperature was selected according to our previous study. The results have been supplements on line 58-68.
Was the freezing time equal in all treatments?
Thanks so much for your questions. The freezing time in all treatments was 24 h as shown on line 91. Therefore, the freezing time was equal in all treatments.
Figure one. The size of the samples seems to be different between each other. How did you measure the localized overheating?
Thanks for your careful review. The size of the samples was same. It seems different just because the photos were taken at different angles and distances. The localized overheating was visual observed. When we saw the edges of samples were white, that means localized overheating occurred. For example, when the core temperature was > -4 °C, varying degrees of maturation were shown in Fig.1.
2. Thawing methods: The bags had holes… so water was in contact with meat when thawing in some treatments?
Thanks so much for your careful review. The bags only for MAT and MWT had holes, otherwise they are unbroken. Just in MWT, water was in contact with meat. The relevant details have been supplemented to clarify this point and shown on line 109-110.
L146 I do not see that TBARS was conducted in the reference cited.
According to your suggestion, the reference has been changed, as shown on line 167.
L191 DSC was not conducted in the reference cited.
According to your suggestion, the reference has been changed, as shown on line 217.
Results and discussion: 3.1. Thawing time and rate L217 “Overall, the thawing rates were higher for the combined methods than the single methods.” I suppose authors are right, but there is no data to prove that. I do not see results of statistical analysis in this section. Did you do it?
Thanks so much for your questions. We have supplemented the data of our previous study as shown on line 246-248. We have performed a statistical analysis of the rates of combined methods and single methods to demonstrate the combined methods could save time. The relevant details were supplemented on line 248-249.
In Figure 2, I can see that MAT and MRT seems to accelerate the speed of the thawing rate at the end of the thawing process. Do you think that this is maybe something interesting to mention?
Thank you very much for the suggestion. The rate-limiting step in the thawing process was passage from the region of maximum ice crystal formation (-5 ~ -1 °C). Thereafter, thawing accelerated. This pattern is consistent with the food freezing. The thawing speed of other treatments in our experiment were fast, so the phenomena were not as clear as MAT and MRT. The relevant clarify has been supplemented on line 235-237.
L233-234: …”and resembled those reported for single thawing treatments [19] but were, in fact, still relatively lower than them”. I cannot found the reference, so I do not understand what single thawing treatments are you referring to. You should explain further.
Thank you very much for the suggestion. We are sorry about that the reference [19] was a Chinese published article. Hence, ‘single thawing treatments’ has been revised to ‘running water thawing’ and the value of thawing loss has been given on line 265.
L235 “Leygonie [3] found” Saying “found” sounds such as it is a research study, but it is a review…
Thank you very much for the suggestion. ‘found’ has been changed into ‘reported’ on line 267.
L235 “accelerated thawing reduced damage”. Could you explain further this, please?
Thank you very much for the suggestion. We have made some corrections to elucidate the point that the damage was thawing loss clearly and shown on line 267-268.
Figure 4 “sarcoplasmic”
Thanks so much for your careful review. ‘Sarciplasmic’ has been revised to ‘Sarcoplasmic’.
L348 Could also the sarcoplasmic protein loss in thawing loss influence these results?
Thanks so much for your careful review. We are sorry that we have made a mistake. The sentence has been rewritten as shown on line 383-386.
Conclusion: L427-428: “In the present study, microwave in combination with other thawing methods effectively reduced thawing time compared to the individual conventional thawing methods. Thawing time could be cut in half by using MAT instead of AT.” This is not a conclusion of the present study. Individual conventional thawing methods were not shown.
Thank you very much for the suggestion. We have deleted these sentences.
…” shorter thawing times” Is shorter compared only to one treatment.
Thank you very much for the suggestion. MAT realizes shorter thawing times compared with AT. However, AT was not done in this experiment. So we have deleted ‘reallizeds shorter thawing times and’.
We hope that the revisions in the manuscript and our accompanying responses will be sufficient to make our manuscript suitable for publication in “Foods”.
Thanks again for your excellent reviewing work. We are looking forward to your reply.
Best regards.
Sincerely,
Ming-Ming Zhu

Reviewer 2 Report
Manuscript ID: foods-661457
Type of manuscript: Article
Title: Physicochemical properties and protein denaturation of pork longissimus dorsi muscle subjected to six microwave-based thawing methods
Authors: Ming-Ming Zhu *, Ze-Yu Peng, Sen Lu, Hong-Ju He, Zhuang-Li Kang, Han-Jun Ma, Sheng-Ming Zhao, Zheng-Rong Wang
Major comments
The manuscript presents valuable results on combination thawing methods using a range of measurements of meat quality. The paper would benefit from more robust discussion of what each of the results means in terms of meat quality.
Also, I recommend checking the references to make sure results in these studies are appropriately used.
In addition, I suggest using a professional proofing service to improve the language in some places of the manuscript.
Section 2.1: please provide more detailed description of how the sample allocation and randomisation were conducted. These are important for assessing the statistical analysis used.
Minor comments
Line 37: replace “would be inevitably occurred” with occur.
Line 47. Reword the sentence to a more neutral statement.
Line 57: replace “based on the previous study” with “and compared with previous studies”.
Line 59: “theoretical foundation”? Please clarify.
Section 2.2.: what were the temperature sensor and measuring protocol?
Line 148: 7.5% w/v?
Line 149: how were the samples filtered?
Line 163: Define TVBN
Line 164: what was the concentration of TCA?
Line 184: what were the assay kits?
Line 298: results in the paper in reference 39 does not support the claim being made here.
Line 303: results in the paper in reference 40 does not support the claim being made here.
Lines 313-314: “All of these meat were thawed by one of the single methods”: so why should the pH result this study be compared to that in the current study?
Line 350: what’s AT?
Figure 4: replace the word solubility on the y-axis and labels by the word concentration.
Author Response
Dear Reviewer,
Thanks for your decision letter on 12th December 2019 about our manuscript.
Manuscript No.: foods-661457.
Title: Physicochemical properties and protein denaturation of pork longissimus dorsi muscle subjected to six microwave-based thawing methods
We appreciated your valuable suggestion and comments. Based on your suggestions and comments, we have thoroughly revised the manuscript and showed in revision mode.
Detailed revisions and responses to the comments are listed below.
The manuscript presents valuable results on combination thawing methods using a range of measurements of meat quality. The paper would benefit from more robust discussion of what each of the results means in terms of meat quality.
Thank you very much for your careful reading, as well as the positive comments and kindly suggestions.
Also, I recommend checking the references to make sure results in these studies are appropriately used.
Thanks so much for your suggestion. The references have been checked and revised to make sure results in these studies are appropriately used.
In addition, I suggest using a professional proofing service to improve the language in some places of the manuscript.
Thanks a lot for your comments on the manuscript. We have engaged an English editorial service. The manuscript has been carefully revised in aspects of wording and phrasing.
Section 2.1: please provide more detailed description of how the sample allocation and randomisation were conducted. These are important for assessing the statistical analysis used.
Thanks so much for your suggestion. We have made some corrections to elucidate the sample allocation and randomisation clearly as shown on line 82-90.
Minor comments
Line 37: replace “would be inevitably occurred” with occur.
Thank you very much for the suggestion. ‘would be inevitably occurred’ has been changed into ‘may occur’ as shown on line 38.
Line 47. Reword the sentence to a more neutral statement.
Thanks so much for your advice. According to your valuable suggestion, the sentence has been written on line 49-50.
replace “based on the previous study” with “and compared with previous studies”.
Thanks a lot for your valuable comments. We have supplemented the results of our previous study in details to clarify ‘why the different treatments used in this study were selected’, as shown on line 58-68. So we deleted ‘based on the previous study’.
Line 59: “theoretical foundation”? Please clarify.
Thanks a lot for your valuable comments. A theoretical basis may be established for the development and optimization of a combination thawing process for application in the meat industry. The unclear expression has been written to elucidate this point clearly as shown on line 74-75.
Section 2.2.: what were the temperature sensor and measuring protocol?
Thank you very much for your questions. The temperature measuring was performed using Testo 160 thermometers. Testo 160 thermometers (Testo Instruments International Trading Co. Ltd., Shanghai, China) recorded temperature changes in the centers of the samples every minute during thawing until the core temperature reached 2 °C. We have made some corrections to elucidate this point clearly as shown on line 121-123.
Line 148: 7.5% w/v?
Thanks for your careful review. ‘w/v’ has been added on line 169.
Line 149: how were the samples filtered?
Thank you very much for your questions. The samples were filtered through Whatman No.1 filter paper and the relevant details have been supplemented on line 171 and line 189.
Line 163: Define TVBN.
According to your suggestion, the definition of TVBN has been supplemented on line 185-187, and related references were inserted accordingly in the reference part.
Line 164: what was the concentration of TCA?
Thank you very much for your questions. Here, the concentration of TCA was 20 g L-1. The relevant data were added as shown on line188.
Line 184: what were the assay kits?
Thank you very much for your questions. The assay kits were total protein quantitative assay kits. We have added ‘a total protein quantitative assay kit’ on line 210.
Line 298: results in the paper in reference 39 does not support the claim being made here.
Thanks for your careful review. We have made a mistake and the sentence has been written on line 333-335. The relevant reference has been changed on line 577-579.
Line 303: results in the paper in reference 40 does not support the claim being made here.
Thanks for your careful review. The unclear expression on line 338-339 might lead to the misunderstanding. We have revised this sentence. The reference 40 referred that ‘The final TBA values obtained in this study did not exceed the flavor threshold (> 1.0 MDA mg/kg) beyond which an undesirable rancid smell and taste of meat products could be expected’ as shown on page 123, section 3.2.2, which support the claim being made here.
Lines 313-314: “All of these meat were thawed by one of the single methods”: so why should the pH result this study be compared to that in the current study?
Thanks for your careful review. We are sorry that we have made a mistake. Here, we just want to clarify ‘there were negligible differences in pH among the various thawed samples (P > 0.05)’. It was not compared and [14] was conducted by using combined thawing methods. Therefore, we deleted this sentence.
Line 350: what’s AT?
Thank you very much for your questions. AT was the brief of air convection thawing. It was occurred on line 68 for the first time.
Figure 4: replace the word solubility on the y-axis and labels by the word concentration.
According to your suggestion, ‘solubility’ has been changed into ‘concentration’.
We hope that the revisions in the manuscript and our accompanying responses will be sufficient to make our manuscript suitable for publication in “Foods”.
Thanks again for your excellent reviewing work. We are looking forward to your reply.
Best regards.
Sincerely,
Ming-Ming Zhu
